# The Use of Ozone Technology to Control Microorganism Growth, Enhance Food Safety and Extend Shelf Life: A Promising Food Decontamination Technology

**DOI:** 10.3390/foods12040814

**Published:** 2023-02-14

**Authors:** Wenya Xue, Joshua Macleod, James Blaxland

**Affiliations:** 1ZERO2FIVE Food Industry Centre, Cardiff Metropolitan University, Cardiff CF5 2YB, UK; 2Cardiff School of Sports and Health Science, Cardiff Metropolitan University, Cardiff CF5 2YB, UK

**Keywords:** ozone, microorganism, food preservation, food safety, food industry, hurdle technology

## Abstract

The need for microorganism control in the food industry has promoted research in food processing technologies. Ozone is considered to be a promising food preserving technique and has gained great interest due to its strong oxidative properties and significant antimicrobial efficiency, and because its decomposition leaves no residues in foods. In this ozone technology review, the properties and the oxidation potential of ozone, and the intrinsic and extrinsic factors that affect the microorganism inactivation efficiency of both gaseous and aqueous ozone, are explained, as well as the mechanisms of ozone inactivation of foodborne pathogenic bacteria, fungi, mould, and biofilms. This review focuses on the latest scientific studies on the effects of ozone in controlling microorganism growth, maintaining food appearance and sensorial organoleptic qualities, assuring nutrient contents, enhancing the quality of food, and extending food shelf life, e.g., vegetables, fruits, meat, and grain products. The multifunctionality effects of ozone in food processing, in both gaseous and aqueous form, have promoted its use in the food industries to meet the increased consumer preference for a healthy diet and ready-to-eat products, although ozone may present undesirable effects on physicochemical characteristics on certain food products at high concentrations. The combined uses of ozone and other techniques (hurdle technology) have shown a promotive future in food processing. It can be concluded from this review that the application of ozone technology upon food requires increased research; specifically, the use of treatment conditions such as concentration and humidity for food and surface decontamination.

## 1. Introduction

There is an increasing need to improve the quality of food and food safety in the industry [1]. Food with a high-nutritional value, safe food additives, fewer production processes, and longer shelf lives and, indeed, that are free from pathogenic microorganisms, represents some of the requirements of both consumers and the food industry [2].

Manufactures have increasingly used a range of preservatives and decontamination methods to control the growth of microorganisms in food and extend its shelf life. Many conventional chemical preservatives/treatments have been evaluated and used, such as the addition of butylated hydroxyanisole (BHA) [3,4], butylated hydroxyltoluene (BHT) [4,5], and tert-butylhydroquinone (TBHQ) [4,6], and for the disinfection, the application of chlorine [7], peracetic acid [8], electrolyzed water [9], and hydrogen peroxide [10]. Other food processing methods including thermal and non-thermal food processing (pulse electric fields, high-pressure processing, pulsed light, ultrasound technology, ionizing radiation, and ozone) have also been reviewed and tested [11,12,13]. The use of chemicals in foods, for example, sodium hypochlorite (NaOCl) at (50–200 mg/L), in disinfecting washing and spraying waters of fresh-cut food, e.g., fruits and vegetables, have long been accepted to reduce microbial populations [14,15]. Nevertheless, many reports have mentioned that the use of chemical preservatives/treatments may harm human health and induce toxic effects in cells because of the disinfectant residues, by-products, and oxidative stress on cell membranes [15,16,17]. The findings of chlorine reactions with natural organic matter in water have been reported to lead to the production of more than 300 different types of by-products such as chlorite, chloride, free available chlorine, and chlorate [18], and there have also been reports of the potential relationship between the exposure to these by-products and toxic effects and the development of adverse outcomes, particularly the cancers of vital organs in human beings [18,19]. 

Ozone is a promising alternative food preserving technique and has gained great interest in the food industry due to its auto-decomposition, rapid action, and strong oxidative properties to produce oxygen, and, most importantly, it leaves no residues in foods from its decomposition and return back to oxygen [2,20,21,22].

Ozone (O_3_; 48 g/mol) acts as a strong disinfectant due to the three oxygen atoms molecule (Table 1). Ozone is formed by the addition of a free radical of oxygen to molecular oxygen. The three oxygen atoms in ozone are arranged at an obtuse angle; the included angle of the two attached oxygen atoms to the central oxygen atom is approximately 116°8’ and the bond length is 1.278 Å. The main zone physicochemical properties are presented in Table 1 [2,21,23,24].

Compared to other oxidising agents, e.g., hydrogen peroxide (1.78 V), chlorine gas (1.36 V), and oxygen (1.23 V), ozone has the potential to react with other substances due to its high oxidising potential (2.07 V; Table 2), e.g., pesticide residue in grain products [22], toxic and micropollutants in waste water [25]. Ozone can also inactivate microorganisms such as fungus [26] and yeasts [27]; pathogenic bacteria such as *Escherichia coli*, *Salmonella typhimurium*, and *Listeria monocytogenes* [28,29]; and viruses and protozoa [30]. The inactivation mechanisms of ozone against microorganisms have been previously discussed, including the ozone penetration of the cells, attacking cell membrane constituents, inactivation of the enzymes, and degradation of the genetic materials of gDNA and total RNA. These activities eventually lead to the leakage of cellular contents and cell lysis [2,29,31].

Ozone was first discovered and observed in 1839 by Schönbein; the gaseous ozone was produced from the electrolysed water [32]. As an antimicrobial agent, ozone has been commercially used for drinking water processing since 1906 in Nice, France [23]. In the U.S. in 1980, ozone was “generally recognised as safe” (GRAS) for bottled water decontamination under specified conditions, which included a maximum ozone dosage of 0.4 mg/L, over 4 min contact time, and that the water to be treated must meet the potable water requirements of the U.S. Environmental Protection Agency [33]. Ozone was considered as GRAS for direct food contact by a panel of experts in 1997 that was requested by the Energy Power Research Institute (EPRI) [32]. In June of 2001, the U.S. Food and Drug Administration (FDA) approved the application of ozone in the treatment, storage, and processing of foods, including meat and poultry [34].

Ozone is generally used in two forms: gaseous and aqueous. There have been a number of previous reviews regarding the application of both gaseous and aqueous ozone in the food industry [21,22,35]. This review aims to collect and summarize all the main factors that influence ozone disinfection efficacy and the papers included in the bibliographical reviews are mostly recent ozone research that have not been reviewed in previous studies in pathogenic microorganism growth control in food processing and preservation. We also explored the potential combined use of ozone technology and other food processing methods that can enhance food safety and extend the shelf life of food products.

## 2. Factors Affecting Microorganism Inactivation Efficiency of Ozone Technology

Depending on the application, ozone is generally employed at a range of concentrations either in its gaseous or aqueous form. As an example, gaseous ozone is produced continuously or periodically to process the harvested product and alter the storage atmosphere [20]. Aqueous ozone is generally applied instantly after the food harvest or during the food washing step [36,37]. In the washing process, the food products can be washed in ozone-dissolved water through different ways, such as: spraying, rinsing, or dipping [38,39,40]. In practice, the efficacy of both ozone phases in the inactivation of microorganisms can be affected by intrinsic and extrinsic factors as previously described [2,41].

### 2.1. Intrinsic Factors

The microbiology intrinsic factors are (1) the microbial load [42], (2) the characteristics of different microbial strains [43,44], (3) physiological states of the microorganism cells [45], and (4) natural or artificially inoculated microorganisms [44]. As demonstrated by Alwi and Ali, the three types of the microorganisms (*E. coli* O157, *S. typhimurium*, and *L. monocytogenes*) react differently to ozone treatment [29]. Alwi and Ali found that gaseous ozone was most effective against *L. monocytogenes* followed by *E. coli* O157 and *S. typhimurium*. The variation in ozone effectiveness against the three types of microorganisms was clearly observed in the treatment with 0.1 ppm ozone for 3 h, which reduced 93.7% of *L. monocytogenes* but only reduced 63.0% and 15.7% of *E. coli* and *S. typhimurium*, respectively [29]. Gibson et al. also observed that multiple comparisons among microorganism type indicate a significantly greater log reduction of *Listeria innocua* (4.7 logs) when compared to *E. coli* (4.2 logs) after ozone sensitisation [46]. Meanwhile, the age and population size of microorganisms can impact their susceptibility to ozone inactivation [2,46]. In Yesil et al.’s study, the effectiveness of gaseous ozone against spinach leaves was significantly affected by pathogen loads comparable to those found in naturally contaminated fresh produce; the efficacy decreased as the inoculum level increased [47].

Other important intrinsic factors are food-property-related, such as (1) the type of food (e.g., fruit, vegetable, meat, and grain), (2) the characteristics of the food surface (e.g. surface size, intact surface, cracks, crevices, hydrophobicity, and texture), and (3) the food weight [43,48] and (4) water activity (aW) of the product itself [49]. The application of ozone for decontamination produces prospective effective results with a low ozone volume requirement on the smooth surface of the products [2], such as apples [50], tomatoes [43], and green peppers [51]. When the products’ surface is more complicated with a high roughness and porosity, such as meat surface, the complete microbial inactivation requires a higher ozone concentration in comparison to smoother surfaces [52]. The aW of a food is the ratio between the vapor pressure of the food itself when in a completely undisturbed balance with the surrounding air media, and the vapor pressure of distilled water under identical conditions. Water activity is a significant factor in relation to the efficacy of the food processing of ozone technology [53]. Wu and colleagues observed the enhancement of gaseous ozone fungicidal efficacy for stored wheat [54]. When the aW of wheat was at 0.80, which means the vapor pressure is 80 percent of that of pure water, after 5 min of gaseous ozone treatment, 30.1% of the spores survived. However, the number of surviving spores was reduced to only 3.1% when the wheat aW increased to 0.90. In the case of 5 min of continuous ozone supply followed by a 30 min holding period, 26.5% of the fungal spores survived at 0.80 aW of wheat, whereas no spores survived at 0.90 aW [54].

### 2.2. Extrinsic Factors

The extrinsic factors were generally ozone treatment factors such as temperature [55], contact time [36,56], ozone concentrations [38,57,58], and application methods (e.g., gaseous ozone treatment, aqueous ozone washing, or combination with other techniques) [35]. It has been observed in previous research that an increase in ozone concentration and microbial exposure time can increase ozone’s antibacterial activity [38,57,58]; these effects have been shown to plateau, which may due to the interference of non-viable microorganism cells or food structures that have been hypothesised to have a protective effect from the oxidative action of ozone [59]. It is also important to distinguish between the concentration of applied ozone and residual ozone necessary for effective antimicrobial control [41]. Thus, it is necessary to monitor ozone availability during the treatment. In Achen and Yousef’s study, bubbled ozone during food processing was observed to be more effective than dipping the food in pre-ozone-bubbled water [50]. Ozone application may also include its use in combination with new technological approaches to enhance the disinfection effectiveness, enhance food safety, and extend shelf life [2]. The emerging combination technologies for potential disinfection in the food industry are in view of the concept of “hurdle technology”. The hurdle technology approach can simultaneously reduce the loss of nutritional contents, sensory quality, and overall processing time [60,61].

As for factors related to water properties for ozone in its aqueous phase, pH [62], organic matter [45], pressure [2], and flow rate [63,64] have been shown to be essential to maximising the oxidising effect of aqueous ozone. The strong disinfecting properties of aqueous ozone are associated with the free radicals that disintegrate the microbial cell walls due to the induced oxidative stress [1]. The efficacy of ozone can be affected by organic matter in solutions as ozone can react with them. When the targeted microorganism is suspended in pure water and simple buffers, the ozone inactivation effects can be detected more readily than in complex food systems [65]. On the other hand, the ozone technology in the food processing industry may form undesirable by-products and jeopardise the safety of the final products [66]. The pH and temperature of the aqueous ozone environment is one of the most critical parameters for aqueous ozone as it is sharply associated with the degree of dissociation of ozone [67]. The concentration of ozone has been reported to be stable under acidic conditions and low temperatures; at pH 3.0 and 8 °C, the highest saturation concentrations were obtained at 4.50 and 8.03 mg/L, with initial gas concentrations of 13.3 and 22.3 mg/L, respectively [68].

In the gaseous phase, the essential factors are air qualities, e.g., air relative humidity (RH) and temperature [55]. It has previously been reported that the effectiveness of gaseous ozone in the activation of microorganisms is highly related to the RH [55,69]. The optimum RH of ozone gas is about 90 to 95% [2,70], while the results in Redfern and Verran’s study indicate that 50% RH can enhance the survival of *L. monocytogenes* at three tested temperatures (4, 10, and 21 °C) [55]. Han et al. tested the disinfection effectiveness of ozone of *E. coli* O157:H7 on green peppers, and RH was tested between 60% and 90%. The strongest inactivation effect of ozone gas was more than 80%. The interaction between ozone gas concentration and RH exhibited a significant and synergistic effect [69].

## 3. Ozone against Microorganisms

Ozone in both its gaseous and aqueous forms has been studied against a wide range of microorganisms including foodborne pathogenic bacteria (e.g., *E. coli*, *Salmonella*, and *L. monocytogenes*) [29], fungi (*Alternaria*, *Aspergillus flavus*, and *Aspergillus parasiticus*) [71,72], viruses, protozoa, and bacterial fungal spores, including spores of *Bacillus*, coliform bacteria, *Micrococcus*, *Flavobacterium*, *Alcaligenes*, *Serratia*, *Aspergillus*, and *Penicillium* [73], and, more recently, the inactivation of coronaviruses on food [74]. Inactivation by ozone is a complex process that involves ozone acting upon various cell structures and cell content constituents [75].

### 3.1. Mechanisms of Ozone Inactivation of Microorganisms

One of the primary reasons for the disinfection ability of ozone is its oxidation-reduction potential (2.08 eV) and increasement of intracellular reactive oxygen species (ROS), which are responsible for bacterial cell lysis and detrimental effect in nucleic acid [76,77]. The major target of the ozonisation treatment is the cell wall, which under stress leads to the leakage of intracellular content as shown in Figure 1. The oxidation of membrane glycoproteins and/or glycolipids has also been shown to occur [78]. Ozone has also been shown to destruct DNA because of the oxidation of double bonds by singlet oxygen [79]. Previous studies concluded two possible primary mechanisms of microorganism inactivation by ozone treatment. The first one includes the ozone exposure oxidation of sulfhydryl groups and amino acids of peptides, proteins, and enzymes to produce smaller peptides, whereas another mechanism involves the oxidation of polyunsaturated fatty acids to acid peroxides [35,66].

Interestingly, previous studies have shown contradictory opinions of the susceptibility of bacteria in ozone exposure. In a study by Moore and colleagues, Gram-negative bacteria were observed to be more sensitive to gaseous ozone (2 ppm for 4 h) than Gram-positive organisms [80]. Rangel et al. also confirmed the effectiveness of gaseous ozone exposure considerably reduced the cell viability of pathogenic Gram-negative bacteria including *E. coli* (30%), *Pseudomonas aeruginosa* (25%), and *Acinetobacter baumannii* (15%) [77], while Cullen and colleagues found that aqueous ozone use in processed fruit juice presented opposite results [81]. Gram-negative bacteria (*E. coli*) appear to be more resistant than Gram-positive bacteria. The differences of the results may be due to the structure differences of the tested strains and the ozone application methods [81]. Cullen et al. reported that they did not observe microbial resistance to ozone treatment, which may be due to the mechanism of ozone action, which destroys the microorganism through cell lysis [81].

### 3.2. Ozone Reaction against Fungi and Mould

Fungal or mould contamination of food is an important aspect in determining food quality and shelf life, with both qualitative and quantitative losses reported due to microbes [82]. Fungal and mould growth leads to the release of secondary metabolites known as mycotoxins, which are dangerous to human and animal health (carcinogenic, teratogenic, and immunosuppressive properties and cause several physiological disorders both in humans and animals) [83]. Ozone has been effectively used to control fungal growth and reduce mycotoxin contamination [84]. Savi and Scussel observed that gaseous ozone (60 μmol/mol for 40, 60, 90, and 120 min) exposure efficiently inhibited the growth of *Fusarium graminearum* and *Penicillium citrinum* [84]. Ozone exposure was found to inhibit conidia germination, and caused hyphae morphological alterations that led to hyphae death and ROS production of the fungi species from *Aspergillus*, *Fusarium*, and *Penicillium* genera [84,85]. Beber-Rodrigues et al. observed that fungi species (*Acremonium*, *Alternaria*, *Aureobasidium*, *Aspergillus*, and *Penicillium*) showed different susceptibilities against gaseous ozone [86]. Ozone disinfection mechanisms can be related to fungi cell metabolism alterations, which lead to apoptosis and oxidative stress, and proved to be effective in controlling toxigenic fungal development [84].

### 3.3. Ozone against Biofilms

Ozone has a promising application in biofilms as it has been shown to deplete the reactive biomass components found within biofilms [75]. Tachikawa et al. observed that aqueous ozone (0.9–3.2 mg/L) treatment is an effective biocide against *Pseudomonas fluorescens* and *P. aeruginosa* biofilms. Nevertheless, the effective concentrations of aqueous ozone for biofilm disinfection may vary with the cell density and structure variance of the biofilms [87]. In a study by Marino et al., aqueous and gaseous ozone were tested against biofilms of three microorganisms (*P. fluorescens*, *Staphylococcus aureus*, and *L. monocytogenes*) [88]. Aqueous ozone under static conditions and for 20 min exposure time resulted in an estimated viability log reduction (between 1.61 and 2.14) of all three microorganisms’ biofilms. Higher log reduction values (3.26–5.23) were observed for biofilms treated in dynamic conditions, which were biofilm-build-coupons maintained under a flow of ozonated water. *S. aureus* was the most sensitive species to aqueous ozone under these conditions. Gaseous ozone at low concentrations (up to 0.2 ppm) reduced 2.01–2.46 log of three microorganisms’ biofilms after 60 min, while at the highest concentrations, it showed a complete inactivation (<10 CFU/cm^2^) of the *L. monocytogenes* biofilms, and the log reductions of 5.51 of *P. fluorescens* and 4.72 of *S. aureus* biofilms were observed [88]. The results indicated that ozone exposure was effective in inactivating microorganisms and could remove exopolysaccharides in the biofilm matrices [75].

## 4. Use of Ozone in Food Preservation and Processing

Ozone has been shown to leave no residue and does not form harmful/carcinogenic by-products on the treated produce [2]. For these reasons, ozone has gained increasing commercial interest for controlling microbial safety and promoting the shelf life of food [21]. In this review, we approach the microbial growth control, and the physical, chemical, and nutritional property effects of fruits, vegetables, meat, and grain processed by ozone. The other goal is to review the combination techniques that can synergistically be applied with ozone treatment in the food industry.

### 4.1. Effects of Ozone in Fruit and Vegetable Processing

Fruits and vegetables are highly consumed in daily life but are susceptible to pathogenic and spoilage-causing microorganisms including bacteria, members of fungi, yeasts, and moulds. Several well-characterised food pathogenic microorganisms were reviewed in a study by Alegbeleye et al. [89]; these contaminants can cause bacterial soft rot, tuber soft rot, yellow lesion, and spoilage of many vegetables and fruits. In addition, contaminated fruits and vegetables can act as vehicles for the transmission of human pathogens such as *E. coli* O26, O111, and O157 [90], *S. typhimurium* [91], and *L. monocytogenes* [92]. The association of *Salmonella* with fresh fruit and vegetables appears to be serovar-specific involving flagella, curli, cellulose, and O antigen capsule [93]. In 2007, the proportion of fruit and vegetable samples that yielded *Salmonella* in the prevalence studies in the UK, Ireland, and Germany ranged from 0.1% to 2.3%, with pre-cut products having some of the highest proportions contaminated [93,94].

Gaseous ozone is regularly used for microbial safety control and postharvest treatment of either intact or fresh-cut fruits and vegetables during storage. Aqueous ozone is mostly applied to wash or rinse the fruits and vegetables to control their physicochemical characteristics and microbiological qualities. A review of studies is presented in Table 3, with the focal point on the ozone treatment effects in microbiology inactivation and the physical, chemical, and nutritional quality aspects of fresh fruits and vegetables.

Gaseous ozone was observed to have an increased oxidation power in respect to increased concentrations [29]. In Alwi and Ali’s study, ozone decontaminated fresh-cut bell peppers and inactivated *E. coli* O157, *S. typhimurium*, and *L. monocytogenes* [29]. These findings were also corroborated in other studies, e.g., 15.008 mg/m^3^ gaseous ozone significantly reduced the microbial populations (>1 log reduction after 14 and 28 days of treatment) on cantaloupes including both bacteria and fungi, which was not seen in low concentration groups (6.432 and 10.720 mg/m^3^) [57]. Additionally, ozone treatment in a time-dependent manner was observed in the studies of both Onopiuk and colleagues and Shu and colleagues [43,58]. Roy et al. reported that a periodic exposure to gaseous ozone at a similar condition as one-time exposure was more effective, and successfully inhibited the growth of both bacteria and mould species with at least a 5 log reduction in microbial colonies [95].

Aqueous ozone application has been shown to be effective in decontaminating the surface of fruit and vegetables, thus extending the food’s shelf life, and that of the ready-to-eat and fresh-cut products they are employed in [2,35]. Botondi et al. concluded that aqueous ozone tends to be more effective in decontaminating intact products than gaseous ozone [96]. Th review results shown in Table 3 prove the effectiveness of aqueous ozone in the decontamination of shredded, fresh-cut, and peeled vegetables [36,38,39]. Ummat and colleagues reported that 2.4 mg/L aqueous ozone application significantly reduced the microbial load on shredded green bell peppers, after exposure for 5 min when stored in polypropylene packages at 5 ± 0.5 °C and 85% ± 5% RH. The authors found that this treatment prolonged the shelf life of the food and it maintained its organoleptic properties for up to 14 days, which was 6 days longer than the control samples [39]. Liu et al. observed significant inhibition effects (*p* < 0.05) of aqueous ozone against not only aerobic bacteria but also coliforms and yeasts during storage; again, the authors found that ozone treatment extended the shelf life of the tested samples [36]. Aslam et al. confirmed the ozonisation treatment of aqueous ozone could extend the shelf life of peeled onion by 14% as compared to water washing, and the authors highlighted that the ozone concentration, exposure time, and aqueous pH factors may all affect the sanitizing potential of ozone [38].

It is important to note that the ozone disinfection process could cause other undesirable changes in quality parameters. Visual and physical aspects such as colour and firmness are important criterion that have an influence on consumer purchases [97]. The effects of ozone technology in the reviewed studies on physical quality parameters, and chemical and nutritional qualities are also described in Table 3. Gaseous ozone at high concentrations (15.008 mg/m^3^) showed a significantly reduced respiration rate and ethylene production rate when compared with controls; additionally, the firmness (at 14 days and 42 days), pectin content (at 28–42 days), titratable acidity (at 14–42 days), sarcocarp, and exocarp (at 14–42 days) of the fruit were also significantly higher than the control group [57]. In aqueous-ozone-treated products, there was no observed negative effect on quality attributes, and a better retention of ascorbic acid, firmness, colour, and overall acceptability during storage as compared to the control samples was found [38,39]. It was also observed that aqueous ozone treatment could remove many pesticides, such as trichlorfon, chlorpyrifos, methomyl, dichlorvos, and omethoate [36].

### 4.2. Effects of Ozone in Meat Products’ Processing

Concerning meat processing, a compilation of the research studies on gaseous- and aqueous-ozone-treated meat and poultry is shown in Table 4, with a focal point on the microorganism inactivation effects of ozone technology and their quality influence.

Jaksch and colleagues evaluated the microbiological inactivation effects of gaseous ozone on pork meat, although the high concentration of ozone was effective at inhibiting and thus reducing physiological activities, but it was not observed to be effective enough to produce a lethal effect on microorganisms present in meat over the 46–49 h analysis time. However, the authors mentioned that the possible reason for the high microbial counts of the ozone-treated samples may be due to the long incubation period after the treatment (46 and 49 h) under non-sterile conditions. Therefore, further studies need to be conducted with increased sample amounts to allow accuracy in the microbial count rate [52]. Gaseous ozone effects at refrigeration temperatures between 0 and 4 °C caused a total inactivation of *E. coli* in culture media [98]. On beef samples, the microbial inhibition was observed at 154 × 10^−6^ kg/m^3^ (72 ppm) at 0 °C and after 24 h, with 0.7 and 2.0 log decreases in *E. coli* and total aerobic mesophilic heterotrophic microorganism (AMHM) counts, respectively. Nevertheless, gaseous ozone exposure at these conditions resulted in observed unacceptable quality aspect parameters in both the surface colour and lipid oxidation of these beef samples. Researchers also reported the antibacterial activity in other ozone treatment conditions; Coll Cárdenas and colleagues reported ozone exposure for 3 h and at both 0 °C and 4 °C, which reduced 0.5 log of AMHM and 0.6–1.0 log of the *E. coli*, without changing the colour or producing rancidity in the tested beef samples [98]. It was also observed that gaseous ozone treatment at 0.01 kg/m^3^ for up to 8 h on turkey meat resulted in 2.9, 2.3, and 1.9 log reductions in total aerobic mesophilic bacteria (AMB), *Enterobacteriaceae*, and yeast-mould, respectively [59]. However, the authors reported undesirable effects on the physicochemical characteristics of turkey breast meat [59]. The high oxidizing potential of ozone can induce alterations in meat products by acting on lipids and proteins, which may be because of the ozone-induced increasement in the thiobarbituric acid reactive substance’s values of turkey breast samples. The lighter colour change in the ozone-treated turkey meat sample could be due to partial denaturation of the connective tissue film layer, which is high in proteins [59].

Periodic gaseous ozone application was applied on beef and was shown to reduce the counts of mesophilic bacteria, lactic acid bacteria (LAB), enterobacteria, moulds, and yeasts, and also prolonged beef refrigerated storage life while retaining acceptable organoleptic qualities. As reported in study [99], gaseous ozone exposure at a concentration range of 276 and 283 mg/m ^3^ for more than 5 min pulse exposure duration and more than 5 pulses had a negative effect on beef colour and caused rancidity [99].

The antimicrobial effects of 1% aqueous ozone were evaluated and compared with 200 ppm chlorine dioxide for the disinfection of beef trimmings by Stivarius et al. [40]. They found that prolonged ozone treatment (>15 min) and chlorine dioxide treatment significantly reduced all microorganisms analysed, whereas a 7 min treatment only effectively reduced *S. typhimurium* (0.75 log reduction) and aerobic plate count (0.32 log reduction). All treatments caused ground beef to become lighter in colour (*p* < 0.05); however, the 15 min aqueous treatment was similar (*p* > 0.05) in redness, percentage discoloration, beef odour, and off odour intensities when compared to control samples [40]. Aqueous ozone and electrolyzed water were not able to reduce the initial microbial counts of the beef samples before vacuum packaging [79]. The targeted RNA-based amplicon sequencing identified that before and after the treatments with aqueous ozone and electrolyzed water, *Pseudomonas fragi* was the most frequent species identified, while aqueous ozone treatments failed to reduce the overall presence of this species, but they affected the intra-species distribution of its oligotypes [79].

### 4.3. Effects of Ozone in Grain Products’ Processing

The persistent use of pesticides has been reported to disrupt biological control systems by natural agents [100], leading to outbreaks of insect pests and the widespread development of resistance [101], undesirable effects on non-target organisms [100], and environmental and human health concerns [102]. Methyl bromide, for example, has been withdrawn by more than 15 industrialized nations due to its environmental risks and reported human toxicity [103]. Apart from the wide spectrum of microbial inactivation and fungal growth control in grain products, ozone has also been reported to be effective in the degradation of mycotoxins such as fumonisins, ochratoxin A, aflatoxins (AFs), zearalenone, deoxynivalenol (DON), citrinin (CTR), and patulin [104]. In Table 5, a summary of ozone treatments for microbiological control, insect management, and mycotoxin degradation is presented.

Ozone as a fumigant is reported to kill stored-grain insects such as Tribolium castaneum, Rhyzopertha dominica, Oryzaephilus surinamensis, Sitophilus oryzae, and Ephestia elutella [109,110]. Gaseous ozone was found to cause the complete mortality of E. kuehniella adults, pupae, and larvae, but only 62.5% of the eggs were killed. Nevertheless, ozone was not reported to be as effective in *T. confusum* management, with 4.2 to 14.1% mortality of adults, pupae, and eggs observed, and only larvae presented a high mortality (74%) [105]. The studies also analysed flush ozone treatment on the top and bottom of stored wheat products and observed almost 100% mortality of all life stages of *E. kuehniella* placed in the top position of wheat, whereas the eggs of *E. kuehniella* placed in the bottom position were found to be resistant.

For *T. confusum*, larvae placed in the bottom position were found to be susceptible to ozone treatment, whereas eggs, pupae, and adults all survived [105]. It was also found that a range of ozone concentrations (e.g., 13 and 21 mg/L) and exposure times (24, 48, 72, and 96 h) effectively controlled aflatoxin-producing species (e.g., *A. flavus*, *A. parasiticus*, and *P. citrinum*) in studies regarding peanut, maize, wheat grains, and corn [72,106,107,108]. In peanut kernels, ozone also caused the reduction in the concentrations of total AFs (30%) and aflatoxin B1 (25%) [72]. In wheat grains, ozone exposure (60 μmol/mol for 180 min) significantly reduced AFB1 from 231.88 to 12.51 μg/kg, AFB2 from 265.79 to 41.06 μg/kg, and CTR levels from 173.51 to 42.90 μg/kg [107]. Ozone exposure at 60 mg/L and for 480 in 1 kg of corn grits, also presented significant reductions in AFG1 (54.6%,), AFB1 (57.0%), AFG2 (36.1%,), and AFB2 (30.0%) [108].

One of the advantages of gaseous ozone is that excess ozone auto-decomposes rapidly to produce oxygen and thus leaves no residues in grain products. Aqueous ozone has been used in various contaminated grains and was observed to quickly react with mycotoxins e.g., DON and pathogens [111,112]. Although aqueous ozone could effectively disinfect and detoxify grain products, most studies on grain products’ decontamination were carried out by employing gaseous ozone owing to the ease of application and solid nature of the product [37]. On the other hand, high-dose ozone application may lead to the oxidation degradation of the chemical constituents present in grain products and may cause surface oxidation and colour change or the development of undesirable odours [109]; thus, the effect of ozone treatment on grain products may not be universally beneficial.

## 5. Combined Applications of Ozone Treatment and Other Technologies in Food Processing

Ozone was reported to have effective antimicrobial properties in food processing; however, when considering its potential negative effects on organoleptic properties more studies need to be undertaken. Hurdle technology utilizes multiple intervention treatments to provide different barriers for microorganisms to overcome for their survival and proliferation, and at the same time they can have a positive effect on the physical, chemical, and nutritional qualities of the food [113]. A summary of the results on ozone in combination with other technologies (hurdle technology) is provided in Table 6.

Placing hurdle technologies may change the microorganism survival environment, such as changing pH, chlorinated compounds, or oxidizing environments, which may compromise the microbial cell wall integrity, metabolism, or, indeed, both, finally resulting in a lethal or inhibitory environment for microbial survival and proliferation. Multiple antimicrobial intervention treatment combinations utilizing 1% ozone with 5% acetic acid or with 0.5% cetylpyridinium chloride on beef trimmings before grinding proved to be effective in reducing bacterial numbers with little effect on beef colour and odour characteristics [114].

In order to overcome these effects, gaseous ozone in combination with carbon monoxide (CO) in various volume ratios was studied by Lyu et al. [115]. Carbon monoxide used as CO can displace O_2_ from oxymyoglobin once it has been bound and carboxymyoglobin is more stable than deoxymyoglobin. In this study, a combination of 5% ozone and 95% CO for 1.5 h was shown to be the most efficacious on inhibiting microbial growth and preventing lipid oxidation [112].

Cantalejo et al. reported the use of ozone (0.6 ppm for 10 min) and lyophilisation in chicken meat preservation and observed that the combination treatment reduced the microbial load of meat and retained its organoleptic properties as well as extending the shelf life by up to 8 months under refrigerated conditions [116].

Another method involves the use of dual-mode frequency ultrasound irradiation and aqueous ozone (sonozonation) and has been shown to enhance the antimicrobial efficiency of treatments through a multi-damage mechanism [117]. The possible reasons behind these results may be because the high pressure generated during the sonication damaged the cell wall and cell membrane structures, because of the enhanced penetration of ozone into bacterial cells after stimulating the mechanisms that aid the elimination of microbial cells from the surface of the vegetables, or because the combination treatment may speed up the collapse of microorganisms, thereby improving the efficacy of the washing process [117].

An alternative combination is ozone with lactic solutions, which was tested in fresh vegetables and proved to significantly reduce both natural mesophilic bacteria and artificially inoculated *E. coli* from the products’ surface. Additionally, the combination treatment showed no undesirable effects on the sensory qualities of the vegetable products [64]. Another study investigated ultraviolet light, as it can cause DNA damage by absorbing light by purines (200 and 300 nm) and pyrimidines (260 and 265 nm) by blocking bacterial replication [118,119]. Ultraviolet light (UV-C) at a cycle dose of 69 mJ/cm^2^ was used in combination with ozone in the inactivation of microorganisms and to preserve the characteristics in beef meat [118]. This methodology prevented the exponential proliferation of microorganisms without modification of the sensory properties of the product [118].

## 6. Conclusions

Ozone is commonly applied in both gaseous and aqueous forms in the food industry to meet the increased consumer preference for a healthy diet and ready-to-eat products. Studies have been conducted to understand its effect on microorganisms on food surfaces, food contact surfaces, and against storage-product insects. Its efficacy is dependent on several factors, including intrinsic (e.g., microbial load and food product properties) and extrinsic (e.g., ozone treatment settings, water qualities, and decontamination methods) factors. Further studies are required to facilitate enhanced control of foodborne microorganisms, including bacteria, fungi, yeasts, mould, and established biofilms on food products, to understand the interactions between intrinsic and extrinsic factors to enable ozone microbiological inactivation and to understand biochemical reactions and the overall effect on the organoleptic properties of food (Figure 2).

This review concluded that the combination of ozone with other technologies showed a promotive future in food processing due to their potential enhancement in antimicrobial effectiveness and the preservation of the sensory qualities of the products. We conclude that ozonisation alone and in combination with other techniques should be further explored, specifically with regards to biofilm reduction and antimicrobial treatment for a range of food categories.

## Figures and Tables

**Figure 1 foods-12-00814-f001:**
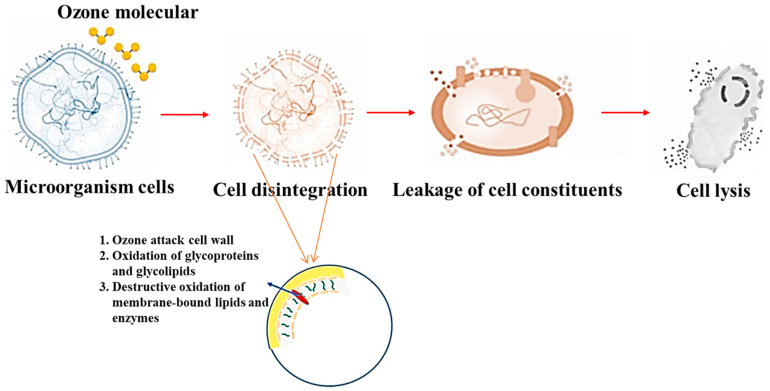
Schematic diagram of microbial inactivation by ozone treatment.

**Figure 2 foods-12-00814-f002:**
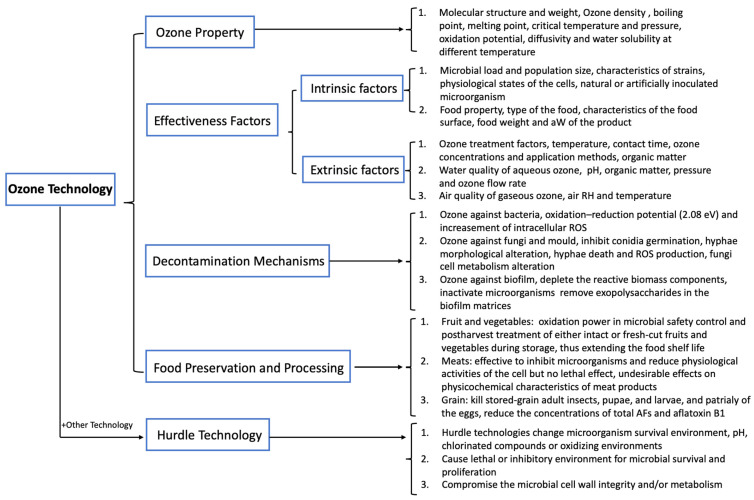
A flowchart to summarise the review studies of ozone technology.

**Table 1 foods-12-00814-t001:** Ozone structure and property [21,23,24].

Parameter	Value
Molecular formula	O_3_
Molecular structure	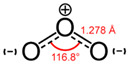
Molecular weight (g/mol)	48
Density (g/L, 1 atm)	2.14
Boiling point (°C, 1 atm)	−111.9
Melting point (°C, 1 atm)	−192.6
Critical temperature (°C, 1 atm)	−12.1
Critical pressure (atm)	54.6
Oxidation potential (V)	−2.07
Diffusivity (20 °C)	1.79 × 10^−9^ m^2^/s (liquid form), 1.46 × 10^−5^ (gaseous form)
Solubility in water at 0 °C (L/L)	0.640
Solubility in water at 15 °C (L/L)	0.456
Solubility in water at 27 °C (L/L)	0.270
Solubility in water at 40 °C (L/L)	0.112
Solubility in water at 60 °C (L/L)	0.000

**Table 2 foods-12-00814-t002:** Oxidising agents and their oxidation potential [23].

Oxidising Agent	Oxidising Potential (V)
Fluorine	3.06
Ozone	2.07
Hydrogen peroxide	1.78
Permanganate	1.67
Chlorine dioxide	1.50
Hypochlorous acid	1.49
Chlorine gas	1.36
Oxygen	1.23

**Table 3 foods-12-00814-t003:** Overview of ozone technology use in microbiological inactivation and its effects on the safety characteristics of vegetables and fruits.

Ozone Application and Conservation Conditions	Produce and Targets	Ozone Treatment Effects in Microbiology	Ozone Treatment Effects on Physical, Chemical, and Nutritional Qualities	References
Gaseous ozone at 0 (control), 1, 3, 5, 7, and 9 ppm; 0.5, 3, 6, and 24 h; 18–20 °C; 95% RH.	Bacterial population change after ozone treatment on fresh-cut bell pepper.	Ozone at 9 ppm, for 6 h, reduced colony counts by 2.89, 2.56, and 3.06 log for *E. coli* O157, *S. Typhimurium*, and *L. monocytogenes*, respectively.	/	[29]
Gaseous ozone at 6.432, 10.720, and 15.008 mg/m^3^; 1 h; weekly occurred. Samples were put on ice by air and were processed immediately at 4 °C after arrival for 42 days.	Microbial safety and postharvest quality of cantaloupes.	Ozone failed to reduce the microbial populations at low concentrations; 15.008 mg/m^3^ ozone effectively reduces the microbial populations and can inhibit most of the bacteria and fungi growth.	The respiration rate and ethylene production rate were significantly lower after 15.008 mg/m^3^ treatment when compared with control and other groups; other factors, e.g., firmness, pectin content, titratable acidity, sarcocarp, and exocarp were significantly higher.	[57]
Gaseous ozone at 0.9 and 2.5 mg/L; 30- and 120 min; 95% RH; up to 15 days; 12 ± 1 °C.	Microbiological properties and health-related properties of Rapanui tomatoes.	Ozonised samples showed lower total amount of yeasts and moulds at 0. Ozone caused a significant reduction in yeast and mould content at day 5, 10, and 15. Ozone at 2.5 mg/L for 120 min was the most effective in bacteria inactivation.	Treatment with ozone increased the content of total soluble solids and reduced titratable acidity and maintained the total flavonoid, lycopene, total antioxidant activity, and total carotenoid content.	[58]
Gaseous ozone at 126–136 ppm; 3 min and 15 min. Ozone was produced by the dielectric barrier discharge generator.	Combinations of spoiled green beans, grape tomatoes, lettuce, and strawberries and *Salmonella enterica*.	Ozone exposure (126–136 ppm, 3 min and 15 min) results in 1 and 4 log reduction, respectively, in food pathogens. Periodic ozone exposure (3 min per day) result in a >5 log reduction of both bacteria and mould species.	/	[95]
Gaseous ozone at 1, 2, and 3 μg/g; 1, 2, and 3 h. Fruit samples were placed in sterile plastic bags and incubatedovernight at 4 °C.	*E. coli* and *L. monocytogenes* survival on tomato.	Ozone insignificantly reduced *E. coli* on tomato; ozone at 3 μg/g caused significant bacteria reduction in a time-dependent manner. For *L. monocytogenes*, 2 μg/g ozone caused significant bacterial reduction with short-duration exposure (1 h).	/	[43]
Aqueous ozone at 1, 1.4, 2, 2.4, and 3 mg/L; 1, 3, and 5 min. Samples were stored at 5 ± 2 °C; 85% ± 5% RH, without any initial gas injection for 16 days.	Physicochemical characteristics, microbiological qualities, and overall acceptability of shredded green bell pepper.	Ozone (>2.4 mg/L) treatments with higher durations significantly reduced the microbial load.	Ozone treatment led to better retention of ascorbic acid, firmness, colour, and overall acceptability as compared to the control samples. The shelf life was 14 days when treated with 2.4 mg/L ozone for 5 min at 5 ± 0.5 °C.	[39]
Aqueous ozone at 1.4 mg/L; 1, 5, and 10 min. Samples were stored at 4 °C for 12 days.	Pesticide residue on fresh-cut cabbage and the growth rates of aerobic bacteria, coliforms, and yeasts.	Approximately 1.2, 1.5, and 1.6 log reduction of aerobic bacteria; 0.2, 0.5, and 0.8 log reductions of coliforms; 1.1–1.4 log reduction of yeasts and a significant reduction in mould in the 1, 5, and 10 min aqueous ozone groups on day 12.	Ozone stimulated initial respiratory metabolism, reduced ethylene production, and improved the overall quality of the samples. Ozone treatment greatly removes trichlorfon, chlorpyrifos, methomyl, dichlorvos, and omethoate.	[36]
Aqueous ozone concentration at 1–5 mg/L; 2–8 min; aqueous pH 3–5.	Microbial reductions, pyruvate content, colour change, and overall acceptability of peeled onion.	Aqueous ozone at 4.51 mg/L exposed to the onions for 8 min at a pH of 3 provided the optimal microbial load reductions (3.74 logs).	The values of pyruvate content ranged from 0.107 (1 mg/L aqueous ozone for 2 min, pH 4) to 0.131 (3 mg/L aqueous ozone for 8 min, pH 3) μM/mL. Non-significant effect of ozone doses on the colour of the samples.	[38]

**Table 4 foods-12-00814-t004:** Overview of ozone technology use in microbiological inactivation and its effects on the safety characteristics of meat products.

Ozone Application and Conservation Conditions	Produce and Targets	Ozone Treatment Effects in Microbiology	Ozone Treatment Effects on Physical, Chemical, and Nutritional Qualities	References
Gaseous ozone at 100 ppm and 1000 ppm; 10 min. The samples were then stored at 25 °C; 46–49 h.	Microbial control of ozone treatment on pork meat.	Ozone treatment greatly suppressed microbial activity. However, ozone treatment failed to effectively reduce the number of microorganisms over the 46–49 h incubation period.	/	[52]
Gaseous ozone at 154 × 10^−6^ kg/m^3^ (72 ppm); 3 and 24 h; 0 and 4 °C.	Ozone effects on AMHM and *E.coli* counts in culture media and in beef samples. Ozone effects on beef quality properties.	Gaseous-ozone-treated *E. coli* media culture after 3 or 24 h, at 0 °C and 4 °C caused a total inactivation of *E. coli*. The highest microbial inhibition was at 0 °C, 24 h exposure, producing a log decrease of 0.7 and 2.0 in *E. coli* and total AMHM counts, respectively.	Ozone treatment for 3 h and at both 0 °C and 4 °C reduced AMHM and *E. coli* counts, without changing the colour or producing rancidity in beef; 24 h treatments failed to significantly reduce microbial counts without affecting beef surface colour and rancidity.	[98]
Gaseous ozone at 0.01 kg/m^3^; up to 8 h, samples were withdrawn at 2 h intervals; 22.0 ± 0.8 °C; 21.6 ± 0.5% RH.	Ozone effects on AMB and *Enterobacteriaceae* counts, and on physicochemical properties of turkey breast muscle.	Gaseous ozone treatment for 6 and 8 h, reduced up to 3 logs of AMB counts. Ozone reduced around 1.0–1.5 log (2 and 4 h) and 2.3 and 2.0 log (6 and 8 h) *Enterobacteriaceae* counts. The yeast-mould count reductions were 0.9 log (2 h) and 1.7 log (4 h). Longer time treatments showed no further inactivation of yeasts and moulds.	Ozone increased carbonyl contents and thiobarbituric acid reactive substances. Ozone caused significant colour and pH value change in the samples. Both water holding capacity and cooking yield of treated samples increased significantly.	[59]
Gaseous ozone 218 mg/m^3^; A: 2 min ozone pulses + 30 min no ozone intervals, for 3 h in total; B: 2 min ozone pulses + 30 min intervals no ozone, for 5 h in total; C: Repeated sample B after 24 h; D: Gaseous ozone 276–283 mg/m^3^. pulses were 5, 10, 20, and 40 min + 30 min no ozone intervals, for 5 h in total. Treatment D (5 min ozone pulse; D5) samples were stored at 4 ± 0.5 °C. D5 samples had repeat inoculation with *L. monocytogenes*; 4 ± 0.5 °C and 10 ± 0.5°C.	Ozone effects on the physicochemical characteristics and food safety of beef.	In A, B, and C, heterotrophic microbial count reductions were between 0.5 and 2 logs. In D, all microorganisms > 1 log reduction. Ozonation intensity showed a significant effect in reducing the counts of mesophilic bacteria, LAB, enterobacteria, moulds, and yeasts. At 4 °C storage, control beef samples (4-day shelf life) showed higher microbial counts than D5 samples (8 day shelf life). D5 showed an immediate around 1 log reduction in *L. monocytogenes* counts. During both 4 °C and 10 °C storages, up to 16 days, *L. monocytogenes* counts in ozonated beef were significantly lower than in control samples.	During refrigerated storage at 4 °C the colour parameters presented no significant differences (*p* > 0.05) when compared with fresh and ozonated beef samples.	[99]
Aqueous ozone at 1% and water bath; 7 and 15 min; 7.2 °C.	Antimicrobial, colour, and odour effects of ozone on ground beef.	Aqueous ozone (15 min) reduced coliforms, *S. typhimurium*, and aerobic plate counts; 7 min treatment effectively reduced *S. typhimurium* and aerobic plate counts.	Aqueous-ozone-treated ground beef became lighter. Minimal effects on colour or odour characteristics by aqueous ozone treatment.	[40]
Aqueous ozone at 6.00 ± 0.25 mg/L. The samples were packed singly in linear low-density polyethylene and vacuum packed and stored at 4 °C.	Ozone effects on the complexity and dynamics of the potential active microbiota of beefsteaks, and their associated volatilome.	Aqueous ozone was not able to reduce the initial microbial counts of the beefsteak samples.	Aqueous ozone was incapable of modifying the microbiota composition, dynamics and the related volatilome to any great extent during chilled vacuum packaging storage.	[79]

**Table 5 foods-12-00814-t005:** Overview of ozone technology use in microbiological and insect species inactivation and its detoxifying effects in grain products.

Ozone Treatment and Conservation Conditions	Produce and Targets	Ozone Treatment Effects on Microbiology, Insect Species and Detoxifying	References
Treatment A: gaseous ozone in a fumigation chamber (3 L) at 13.88 mg/L; 2 h; treatment B: gaseous ozone (13.9 mg/L) flush treatment of 2 kg wheat in 3 L chamber at 30 min intervals with 10 pulses for 5 h in total.	Effectiveness of ozone on the mortality of stored-product insects’ (*Ephestia kuehniella* and *Tribolium confusum*) larvae, pupae, eggs, and adults.	Empty space ozone treatment caused complete mortality of *E. kuehniella* adults, pupae, and larvae, 62.5% of the eggs were killed. Ozone treatment caused low mortality of *T. confusum* adults, pupae, and eggs, ranging from 4.2 to 14.1%, only larvae had a high mortality (74%). Ozone flush treatment caused almost complete mortality of all life stages of *E. kuehniella* placed in the top position of 2 kg wheat, whereas eggs of *E. kuehniella* placed in the bottom position were hard to kill. *T. confusum*, larvae placed in the bottom position were easily killed, eggs, pupae, and adults survived.	[105]
Gaseous ozone at concentrations of 13 and 21 mg/L; 0, 24, 48, 72, and 96 h.	The fungicidal and detoxifying effects of ozone on AFs in peanut kernels.	Ozone at 13 and 21 mg/L effectively controlled the potential aflatoxin-producing species *A. flavus* and *A. parasiticus*. Ozone at 21 mg/L for 96 h effectively controlled total fungi and potentially aflatoxigenic species in peanuts, with a > 3 log (CFU/g) reduction. Ozone also caused a reduction in the percentage of peanuts with internal fungal populations. Ozone-treated kernels at 21 mg/L for 96 h caused a reduction in the concentrations of total AFs and aflatoxin B1.	[72]
Gaseous ozone at rates of 0, 50, 500, 1000, and 15,000 ppm in factorial with moisture contents of 18, 22, and 26% for 1 h, at 0.5 L/min flow rate.	Ozone treatment efficacy of high-moisture maize to reduce the occurrence of fungal infections within kernels during storage	Ozone concentration at 500 and 1000 ppm effectively reduced the presence of *Aspergillus*, *Fusarium* and *Mucor*. *Penicillium* infections decreased with ozone at 1000 and 15,000 ppm. Ozone at 15,000 ppm was necessary to reduce *Rhizopus* infection. Ozone can penetrate the surface of maize kernels to reduce fungal infections during storage.	[106]
Gaseous ozone at 40 and 60 μmol/mol; 30, 60, 120, and 180 min; 25 ± 0.5 °C.	The effectiveness of ozone treatment against *A. flavus* and *P. citrinum* strains’ growth as well as AFs and CTR degradation in wheat grains.	Ozone at 40 and 60 μmol/mol >30 min significantly reduced *A. flavus* and *P. citrinum*. Ozone at 60 μmol/mol, for 180 min, showed 100% growth inhibition of *A. flavus* and *P. citrinum* and significantly reduced AFB1 and AFB2 levels. Ozone at 40 and 60 μmol/mol for 180 min significantly reduced CTR levels.	[107]
Gaseous ozone concentration at 20 to 60 mg/L; 120 to 480 min.	The effects of ozonation to corn grits, including the levels of AFs (B1, B2, G1, and G2), fungal contamination, and total mesophilic count.	Ozone at highest concentration 60 mg/L and 480 min exposure time and 1 kg of corn grits, reached log reductions of 2.04 (*Aspergillus* spp.) and 2.77 (*Fusarium* spp.) in corn grits (CFU/g), total mesophilic counts were reduced to non-detectable levels. After above ozone detoxification, observed greatest reductions were for AFG1, AFB1, AFG2, and AFB2.	[108]

**Table 6 foods-12-00814-t006:** Overview of ozone in combination with other technologies.

Ozone Treatment Conditions	Combination Technologies	Produce and Targets	Ozone Treatment Effects		Combination Effects		References
			Microbiology	Other Qualities	Microbiology	Other Qualities	
Aqueous ozone at 1%; water bath sample at 7.2 °C for 15 min.	5% acetic acid/0.5% cetylpyridinium chloride.	Antimicrobial, colour, and odour effects of ground beef.	Ozone treatment reduced coliforms, *S. typhimurium*, and aerobic plate count.	Samples became lighter; similar redness, percentage discolouration, odour, and off odour intensities as the control	Ozone with 5% acetic and 0.5% cetylpyridinium chloride reduced all bacterial types.	Combination treatments showed little effects on sample colour and odour.	[114]
Gaseous ozone at 2%, 5%, and 10%.	CO modified atmosphere package.	Combination effects on the microbiological, chemical, physical, and sensory characteristics of beef.	Ozone at 5% and 10%, caused the largest reduction in total viable counts on day 0.	The drip loss, metmyoglobin, thiobarbituric acid reactive substances, total volatile basic nitrogen, and pH were significantly lower in >2% ozone-treated samples.	The total viable counts of >2% ozone groups were reduced significantly when compared with the CO only groups.	The combination treatment significantly reduced the drip loss, metmyoglobin, thiobarbituric acid reactive substances, total volatile basic nitrogen, and pH.	[115]
Gaseous ozone at 0.4, 0.6, and 0.72 ppm; 10, 30, 60, and 12 min; 4 ± 0.5 °C; 90 ± 1% RH.	Slow freezing, 20.5 h of primary drying (12 h at 0 °C and 8.5 h at 10 °C) at 30 Pa.	Combination effects on the microbiological load, sensory characteristics, and shelf life of chicken.	Ozone (>0.4 ppm) significantly reduced total aerobic mesphilic bacteria counts, lactic acid bacteria counts throughout 8 months.	Ozone (0.4 ppm, >30 min) increased the aW and humidity; decreased the rehydration of the samples.	The combination reduced the total AMB, the mesophilic, and lactic acid bacteria counts.	The combination significantly reduced the pH values, the aW, and humidity;. increased the maximum force value.	[116]
Aqueous ozone at 0.85 ± 0.2 mg/L; 5, 10, and 15 min.	Ultrasound: mono-mode frequency irradiation, dual-mode frequency irradiation.	Microbial safety and nutritional quality, firmness, bioactive compounds, and antioxidants of cherry tomato.	Ozone reduced the mesophilic bacteria (0.40–0.71 logs) and moulds/yeasts (0.29–0.49 logs).	Ozone slows down the loss in firmness (23.07–24.58%) after 21 days storage. Ozone-treated samples had the lowest electrolyte leakage, less loss in bioactive compound, and increased antioxidant activity.	The dual-mode frequency irradiation with ozone reduced mesophilic bacteria (2.09–3.42 logs) and moulds/yeasts (2.30–3.72 log).	The combinations slowed the maturity process; maintained the bioactive compounds, total soluble solids content, titratable acidity, and pH values; increased the antioxidant activity.	[117]
Ozone (3–9 mg/L) passed into a covered beaker and with sterile water through a sparger.	Lactic acid solution	The removal of microbial and chemical contaminants from fresh vegetables.	Ozonated water at 9 mg/L for 10 min reduced 0.9–2.4 logs of natural microbes and 1.3–2.1 logs of *E. coli* from vegetable samples.	/	Combinations Reduced natural mesophilic bacteria and *E. coli* from tomato, cucumber, carrot, and lettuce.	The combinations showed no effects on the sensory quality of fresh vegetables.	[64]
Aqueous ozone at 0.9 ppm in cycles; a total of 10 sprays for 30 s with an interval of 1 h, total 10 h.	A: UV-C (15 s of) +30 s ozone spraying; 10 cycles; 10 h. B: UV-C alternately applied; 10 cycles; total 10 h.	The effects of treatments in the microorganisms and in preservation of beef meat characteristics.	A significant microbial reduction (*p* < 0.05) was not observed concerning the initial control sample.	/	A: significant reductions in all 10 cycles. Cycles 5 and 8, a 0.7 log reduction of *E. coli*. Initial microbial load was maintained in other cycles. B: significant reduction in microbial load for cycles 2–10.	The action of the combined treatments on the meat showed no effects in the pH, lipid oxidisation, and total protein amount.	[118]

## Data Availability

The data presented in this study are available on request from the corresponding author.

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
