# Peer review of "The Use of Ozone Technology to Control Microorganism Growth, Enhance Food Safety and Extend Shelf Life: A Promising Food Decontamination Technology"

_foods, 2023, doi:10.3390/foods12040814_

Round 1
Reviewer 1 Report
The ozone technology reviewed in the present research is a promising technology. The aim is significant to this field and reader-attracted. But some important parts need to be considered.
1. For the type of review article, the flowchart is necessary to summarize the whole.
2. The logic and expression are not very clear. Such Section 2: Factors affecting microorganism inactivation efficiency of ozone technology, intrinsic factors, or extrinsic factors, point 1, point 2, point 3, etc.
3. The applications of ozone technology in food storage and processing is a good description but the nutrition remaining or appearance enhancing are not mention.
4. The title maybe focus on the industrial applications in food industry, but the limited industry applications in the present form.
Author Response
Thank you for your time and effort in reviewing our paper. We value your feedback on how to improve and have acted on your comments. Please see detailed description below:
- For the type of review article, the flowchart is necessary to summarize the whole.
Yes agreed – we have added a flowchart at the end to summarise the findings of the study. This flowchart will also form the diagram used for MDPI on its website post publication. Thank you.
- The logic and expression are not very clear. Such Section 2: Factors affecting microorganism inactivation efficiency of ozone technology, intrinsic factors, or extrinsic factors, point 1, point 2, point 3, etc.
Agree with your comments here, after revising the manuscript it wasn’t entirely clear, we have therefore added some bullet points and rearranged the text to flow better.
- The applications of ozone technology in food storage and processing is a good description but the nutrition remaining or appearance enhancing are not mentioned.
Yes, excellent. This is an important aspect that was overlooked. We have added detail to this effect in lines 430-433.
- The title maybe focus on the industrial applications in food industry, but the limited industry applications in the present form.
Agreed – we have re-written the title to represent the purpose of the study – Title: “The use of ozone technology to control microorganism growth, enhance food safety and extend shelf life: A promising food decontamination technology”. This review focus on the use of ozone in food decontamination and microbiology control, this can be potentially used in food industry and give an industry setting.
Thank you for all your feedback.
Reviewer 2 Report
Foods
foods-2194148
The use of ozone technology in the food industry to control microorganism growth, enhance food safety and extend shelf life: A promising technology
Dear Editor,
In the paper, the chemical and physical properties of ozone, ozones oxidation potential, and the intrinsic and extrinsic factors that affect microorganism inactivation efficiency of both gaseous and aqueous ozone are explained, as well as the mechanisms of ozone inactivation of foodborne pathogenic bacteria, fungi, mould, and biofilms. The topic is good. The paper has been generally well designed and written. My questions and comments;
- Line 56: Please give the possible reason of this effect!
- Some information, which is given in Table 1, has been also given in the text. Please don’t replicate!
- Give the mechanism behind these activities of ozone in detail. Explain the effect of ozone on the color of the foods!
- Line 353: Please explain the effect of ozone on the lipid oxidation. Because it is very important in terms of meat quality!
Author Response
Thank you for your time and effort in reviewing our paper. We value your feedback on how to improve and have acted on your comments. Please see detailed description below:
- Line 56: Please give the possible reason of this effect!
Yes absolutely, a really good point that we overlooked, thank you - Re: Ozone decomposition and its affecting factors were added at the Section 2.1 and Section 2.2
- Some information, which is given in Table 1, has been also given in the text. Please don’t replicate! Remove text and point to the table
Yes, perfect, and will make the text concise and clear - we have rewritten and re-arranged the review text regarding ozone properties.
- Give the mechanism behind these activities of ozone in detail. Explain the effect of ozone on the colour of the foods
Thank you for highlighting this really important point – we have added more information to reflect the ozone effect on food colour on lines 355-370
- Line 353: Please explain the effect of ozone on the lipid oxidation. Because it is very important in terms of meat quality!
Yes we totally agree and something we have overlooked. We have added more information to lines 355-358
Thank you for all your feedback and time.
Round 2
Reviewer 1 Report
The present manuscript meets the Journal Publication requirements.